# Relationships among COVID-19 phobia, health anxiety, and social relations in women living with HIV in Iran: A path analysis

Fatemeh Aliverdi[1], Zahra Bayat Jozani[2], Nooshin Ghavidel[3], Mostafa Qorbani[4,5]☯*, Nami Mohammadian Khonsari[4], Farima Mohamadi[6], Minoo Mohraz[7], Zohreh Mahmoodi[3]☯*

1 Student Research Committee, Alborz University of Medical Sciences, Karaj, Iran, 2 Department of Reproductive Health and Midwifery, School of Nursing and Midwifery, Tehran University of Medical Sciences, Tehran, Iran, 3 Social Determinants of Health Research Center, Alborz University of Medical Sciences, Karaj, Iran, 4 Non-Communicable Diseases Research Center, Alborz University of Medical Sciences, Karaj, Iran, 5 Endocrinology and Metabolism Research Center, Endocrinology and Metabolism Clinical Sciences Institute, Tehran University of Medical Sciences, Tehran, Iran, 6 Social Determinants of Health Research Center, Shahid Beheshti University of Medical Sciences, Tehran, Iran, 7 Iranian Research Center for HIV/AIDS, Iranian Institute for Reduction of High-Risk Behaviors, Tehran University of Medical Sciences, Tehran, Iran

☯ These authors contributed equally to this work.
* mqorbani1379@yahoo.com (MQ); Zohrehmahmoodi2011@gmail.com (ZM)

**Data Availability Statement:** All relevant data are available on Zenodo: https://doi.org/10.5281/zenodo.7098735.

## Abstract

### Introduction

The COVID-19 pandemic and its consequences have caused fear and anxiety worldwide and imposed a significant physical and psychological burden on people, especially women living with HIV (WLHIV). However, WLHIV were not studied as well as others during the pandemic. Hence, this study aimed to determine the relationships between COVID-19 phobia, health anxiety, and social relations in WLHIV.

### Materials and methods

This cross-sectional study enrolled 300 WLHIV who had records at the Iranian Research Center for HIV/AIDS of Tehran University of Medical Sciences. Data were collected using sociodemographic questionnaire, the fear of COVID-19 scale, the social relations questionnaire, the socioeconomic status scale and the health anxiety inventory. Path-analysis was used to assess the direct and indirct associations between variables.

### Results

Based on the path analysis, among variables that had significant causal relationships with social relations, socioeconomic status (β = -0.14) showed the greatest negative relationship, and health anxiety (β = 0.11) had the strongest positive relationship on the direct path. On the indirect path, fear of COVID-19 (β = 0.049) displayed the greatest positive relationship. The level of education (β = 0.29) was the only variable showing a significant positive relationship with social relations on both direct and indirect paths.

**Funding:** The authors received no specific funding for this work.

**Competing interests:** The authors declare that they have no competing interests.

**Abbreviations:** WLHIV, women living with HIV; AGE, age of participant; PN, partner number; CN, Child Number; HA, Health anxiety; EDU, Education; CPH, COVID-19 phobia; SES, Socioeconomic statues; SR, Social relation; Min, Minimum; Max, Maximum.

## Conclusion

Our result showed that increased fear and health anxiety related to a higher social relations score in WLHIV. Hence, due to their vulnerability, these people require more support and education to adhere to health protocols in future pandemics and similar situations.

## 1. Introduction

The COVID-19 pandemic has affected the psychological status of many people [1]. Based on the Inter-Agency Standing Committee (IASC) report, people are, directly and indirectly, impacted by stressful experiences in this period; and their most prevalent responses to these stressful experiences include fear (of illness, death, loss of livelihood, social isolation, and being quarantined) and fear-related behaviours, e.g., limited social relations, distance from treatment centers (fear of going to a health facility), health anxiety, depression, and stress [2].

Fear is an adaptive feeling needed to cope with potential threats, but excessive fear has negative impacts on the personal (mental health issues and anxiety disorders) and social levels (seclusion, isolation, xenophobia) [3]. Researchers have discussed the pathological fear of COVID-19 (COVID-19 phobia) due to the nature and wide-ranging impacts of the pandemic [4].

Various factors may affect the degree of psychological vulnerability to COVID-19 phobia, including personal variables such as tolerance, lack of trust, vulnerability to the diseases, anxiety, and concerns [4]. Reports suggest that older adults and those with underlying diseases, including HIV, are at greater risk [5]. Hence it seems that the COVID-19 pandemic imposes a more significant physical and psychological burden on women living with HIV (WLHIV) [6, 7], as WLHIV have severe early death anxiety, different disease-related fears, and mental disturbances ranging from indifference and hopelessness to severe reactions such as anxiety and depressive disorders [8].

Health anxiety is a wide-ranging cognitive disorder formed by incorrect perceptions regarding physical changes and symptoms resulting from one's beliefs about illness or health [9]. Some researchers say severe health anxiety can arise from COVID-19 phobia [4]. Almost everyone has experienced some degree of health anxiety, low levels of which are not pathological but rather help people perform and commit to preventive behaviours. However, severe degrees are associated with maladaptive coping behaviours leading to distress, social incompetence (social disability), disrupted job performance, and having a fear of going to health centers (even in situations when it is necessary) or having an obsession with going to the health centers repeatedly without any necessity [10].

In social science, a social relation is any relationship between two or more individuals. People inherently need and thus create opportunities to experience social relations. Social relations also affect mental health, health-related behaviours, and physical health. Studies show that social relations have short- and long-term effects on health that emerge during childhood and throughout one's life [11].

During the COVID-19 pandemic, the World Health Organization (WHO) recommended precautionary measures, including quarantine, limiting social relations by increasing physical distance, wearing a mask when visiting others, and avoiding overcrowding [12]. Due to the importance of this topic, the vulnerability of WLHIV, and the absence of a model examining all the mentioned variables together, especially for this group, the current study aimed to determine relationships among COVID-19 phobia, health anxiety, and social relations in WLHIV via path analysis.

## 2. Materials and methods

### 2.1. Study design

In this cross-sectional study, 300 WLHIV who had records at the Iranian Research Center for HIV/AIDS affiliated to Tehran University of Medical Sciences were included.

### 2.2. Study population

The sample size was calculated according to Maria Pizzirusso et al. [13] study, by considering type I ($\alpha$) and type II ($\beta$) errors of 0.05 and 0.2, respectively, and the correlation (r) of 0.16 for social relations and anxiety, by using the following formula:

Total sample size $[(Z_\alpha+Z_\beta)/C]^2 + 3$

When C = 0.5 $^*$ ln [(1+r)/ (1-r)]

**Inclusion criteria.** Iranian women with records at the Iranian research center for HIV/ AIDS of Tehran University of Medical Sciences, with minimum literacy, absence of mental and physical problems (as reported by the patient/registered in their records) that would preclude them from participation, and no history of psychotropic medications.

**Exclusion criteria.** Returning incomplete questionnaires, migration, and hospitalization due to COVID-19.

### 2.3. Data collection and definition of terms

The data were collected via four questionnaires: The fear of COVID-19 scale [14]; The health anxiety inventory [15]; the social relations questionnaire; the socioeconomic status (SES) scale [16]; as well as a sociodemographic information checklist.

**2.3.1. The fear of COVID-19 scale.** Pakpour, Griffiths, et al. developed the fear of COVID-19 scale in 2020 with seven items. 1-I am most afraid of COVID-19. 2-It makes me uncomfortable to think about COVID-19. 3-My hands become clammy when I think about COVID-19. 4-I am afraid of losing my life because of COVID-19. 5-When watching the news and stories about COVID-19on social media, I become nervous or anxious.6-I cannot sleep because I'm worried about getting COVID-19-19.7-My heart races or palpitates when I think about getting COVID-19 [14]. The responses range from "strongly disagree" (1) to "strongly agree" (5). All items' scores yield a total score ranging from 7–35. The original version has a Cronbach's alpha of 0.82, test-retest coefficient of 0.88, and appropriate validity. In Iran, its reliability was confirmed with a Cronbach's alpha of 0.86 [17]. The scale's reliability was confirmed in the current study with a Cronbach's alpha of 0.84.

**2.3.2. The health anxiety inventory.** The health anxiety inventory was developed by Salkovskis and Warwick (2002) with 18 items to measure health anxiety. Each question consists of a group of four statements describing a person's health or feelings about health and illness over the past six months. There are no right or wrong answers. The questions are scored on a four-point Likert scale (never = 0 to often = 3). Salkovskis reported the test-retest reliability of 0.90 and Cronbach's alpha of 0.70–0.82 [15]. The Persian version of the inventory showed a Cronbach's alpha of 0.75, demonstrating optimal validity [18].

**2.3.3. Social relations questionnaire.** This questionnaire evaluates communication skills and social functioning with others/community and consists of 11 items like "I don't worry about dealing with people, I have competence and ability in social affairs, I am interested in joining a private group, I can be successful in verbal communication with others, Weak social connections can be due to failures caused by the living environment and society", and it's scored on a five-point Likert scale from very low to very high (1–5). The score ranges from 11

to 55. Its reliability was confirmed with a Cronbach's alpha of 0.87 (Mousavi, 2013). The current study confirmed its reliability with a Cronbach's alpha of 0.89.

**2.3.4. Socioeconomic status (SES) scale.** SES consisted of 6 questions, including parental education, income, economic class, and housing status, which are scored based on a Likert scale from 1 to 5, and a total score ranging from 6 to 30. Validity and reliability was approved in Iranwith a Cronbach's alpha of 0.83 (2013) [16].

**2.3.5. Socio-demographic checklist.** Sociodemoghraphic characteristics including age, duration of the disease, the number of children, having sex partners, education, and having health insurance were collected via self-constructed checklist.

## 2.4. Procedure

The study began after obtaining the required permissions and approval from the Ethics Committee of Alborz University of Medical Sciences (IR.ABZUMS.REC.1400.022). The researchers visiting the centre of the behavioural diseases who were identified as eligible participants were briefed about the study's objectives. The eligible participants signed a written informed consent form if they were willing to participate. Due to the COVID-19 pandemic and to adhere to distancing and minimal presence at the center, the questionnaires were sent to those who had Internet access over the Pars Online platform, and they were requested to fill them out in one week. For those who did not have Internet access, a separate room was allocated for filling out the questionnaires. The respondents could ask their questions regarding questionnaire items and resolve any ambiguities by phone for those who used the Internet and in person for those who filled out the questionnaire in the center.

They were all ensured that their data would remain confidential, that participation was not obligatory, and that they would not be deprived of any services if they did not participate.

## 2.5. Statistical analysis

This study examined the fitness of a conceptual model for the relationship between fear of COVID-19, health anxiety, and social relations in WLHIV. Path analysis is an extension of conventional regression that shows each variable's direct and indirect effects on the dependent variables. The results can be used to provide a rational interpretation of the relationships and correlations observed. It can consider a causal modelling technique; it can be performed with cross-sectional data [19]. Direct and indirect effects of independent variables included age, education, child number, SES, partner number, health anxiety, COVID-19 phobia on social relations was assessed using the path model. The results of path model was reported as standardized and unstandardized β coefficient. In the path model, the significance level was set at a T-value >1.96.

Pearson's correlation coefficient was used to assess correlation between variables. P-value less than 0.05 was considered as statistically significant. Data were analyzed in SPSS-25 and Lisrel-8.8.

## 3. Results

Demographic characteristics are presented in Table 1. The participants' mean age was 39.4 ± 7.5 years. Mean (SD, minimum, maximum) of the health anxiety score, the fear of COVID-19 score, and social relations score (35.1 ± 3.80) was 20.6 (7.3, 5,52), 22.4 (5.3,7,35) and 35.1 (3.80, 26, 48) respectively.

Based on Pearson's correlation analysis, among independen variables education had the strongest significant positive correlation (r ≈ 0.37), and the number of children displayed the strongest significant negative correlation (r ≈ -0.26) with social relations (Table 2).

**Table 1. Sociodemographic characteristics of participants.**

| Variable | | Value |
|---|---|---|
| Age (year)[1] | | 39.4 (7.5) |
| HIV duration (year)[1] | | 7.88 (4.98) |
| Education (year)[1] | | 10.07 (3.65) |
| Social-economic status (score)[1] | | 9.5 (2.61) |
| Health anxiety (score)[1] | | 20.6 (7.3) |
| Corona Phobia (score)[1] | | 22.4 (5.3) |
| Social relation (score)[1] | | 35.1 ± 3.8 |
| Covid-19 positive history [2] | | 72 (23.8) |
| Having insurance | | 95 (31.5) |
| Number of children[2] | 0 | 116 (38.4) |
| | 1 | 81 (26.8) |
| | 2 | 60 (19.9) |
| | 3 and more | 45 (14.9) |
| Job[2] | Housekeeper | 83 (27.5) |
| | Worker | 84 (27.8) |
| | Employee | 23 (7.6) |
| | Unemployed | 47 (15.6) |
| | Retired | 11 (3.6) |
| | Self-employed | 54 (17.9) |

[1] are reported as mean (SD)

[2] are reported as number (%)

Based on the path analysis, among variables with significant and causal relationships on the direct path with social relations, socioeconomic status ($\beta$ = -0.27) had a negative relationship, while health anxiety ($\beta$ = 0.75) and HIV duration($\beta$ = 0.51) had a positive relationship. In other words, with a one-score increase in SES, the social relations score decreased, and with a rise in the health anxiety score and duration of HIV, the social relations score increased. On the indirect path, the fear of COVID-19 ($\beta$ = 0.32) had a significant and positive relationship with social relations; in other words, a rise in fear of COVID-19 score was associated with a

**Table 2. The correlation matrix of sociodemographic variables, health anxiety, COVID-19 phobia, and social relations in PLWH.**

| | Age | Partner number | Child number | HIV duration | Health anxiety | SES | Covid-19 phobia | Education | Social relations |
|---|---|---|---|---|---|---|---|---|---|
| Age | 1 | -0.152** | 0.587** | 0.448** | -0.034 | 0.034 | 0.124* | -0.356** | -0.235** |
| Partner number | | 1 | -0.129* | -0.091 | -0.069 | 0.141* | -0.035 | 0.153** | 0.173** |
| Child number | | | 1 | 0.289** | 0.026 | -0.039 | 0.184** | -0.343 | -0.255** |
| HIV duration | | | | 1 | -0.081 | 0.025 | 0.037 | -0.170** | -0.167** |
| Health anxiety | | | | | 1 | -0.009 | 0.424** | -0.013 | -0.108* |
| SES | | | | | | 1 | 0.034 | 0.184** | 0.192** |
| Covid-19 phobia | | | | | | | 1 | -0.1 | -0.033 |
| Education | | | | | | | | 1 | 0.365** |
| Social relations | | | | | | | | | 1 |

**: P 0.01

*: P>0.05, SES: social-economic statuse

**Table 3.  The direct and indirect effects of sociodemographic variables, health anxiety, COVID-19 phobia and social relations in PLWH.**

| | Standardized β coefficient | | | Unstandardized β coefficient | | | R² |
| --- | --- | --- | --- | --- | --- | --- | --- |
| | Direct effects | Indirect effects | Total effect | Direct effects | Indirect effects | Total effect | 0.96 |
| Age | 0. 1* | 0.06 | 0.1* | 0. 5* | 0. 4 | 0.5 * | |
| Education | 0.37* | -0.048* | 0.32* | 0.5* | -0.06* | 0.44* | |
| Child number | 0.13* | -0.02 | 0.13* | 0.5* | -0.09 | 0.5 | |
| SES | -0.27* | -0.015 | -0.27* | -0.5* | -0.025 | -0.5* | |
| Partner number | - | -0.05 | -0.05 | - | -0.145 | -0.145 | |
| Health anxiety | 0.75* | - | 0.75* | 0. 5* | - | 0. 5* | |
| HIV duration | 0.51* | -0.061 | 0.51* | 0.5* | -0.059 | 0.5* | |
| COVID-19 phobia | - | 0.322* | 0.322* | | 0. 3* | 0. 3* | |

*Statistically significant, SES: social economic statuse

rise in social relations score. Level of education was the only variable showing a significant and positive relationship with social relations on both direct and indirect paths (B = 0.32), meaning that a higher level of education was associated with a higher social relations score (Table 3) and Figs 1 and 2.

The model's fitness indices demonstrate its goodness of fit and the reasonably adjusted relationships among the variables (Table 4).

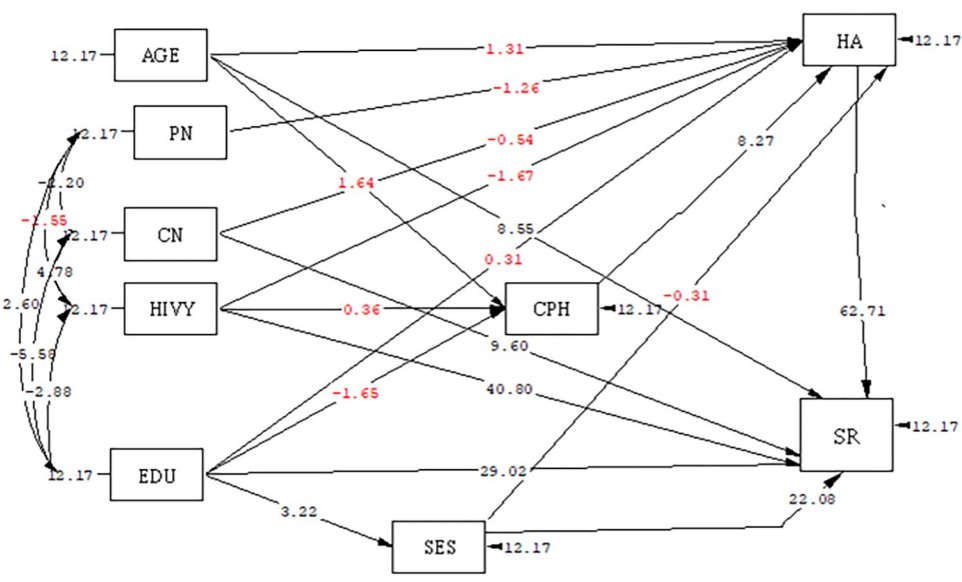

Chi-Square=87836.94, df=9, P-value=0.00000, RMSEA=5.742

**Fig 1. Full empirical path model between health anxiety, COVID-19 phobia, and social relations according to T-value.** (T-value>1.96 is considered as significant). AGE: Age, PN: Partner number, CN: Child number, HA: Health anxiety, HIVY = HIV duration (year), SES: Socioeconomic status; EDU: Education, SR: Social relations; CPH: COVID-19 phobia.

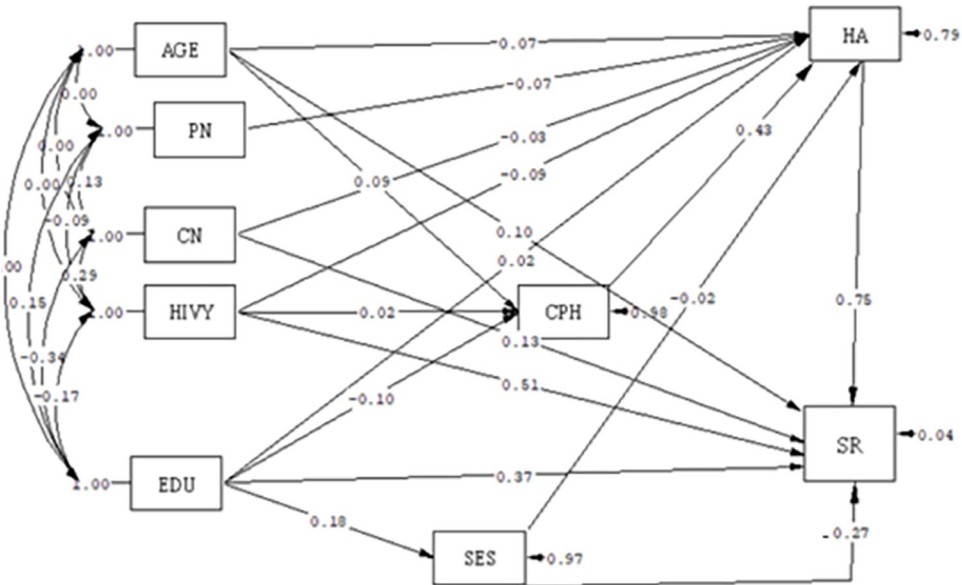

**Fig 2. Full empirical model (Empirical path model between health anxiety, corona phobia, and social relations) according standard B.** AGE = age of the participants, PN = partner number, CN = Child Number, HA = Health anxiety, HIVY = HIV duration, EDU = Education, CPH = COVID-19 phobia, SES = Socioeconomic statues, SR = Social relation.

## 4. Discussion

This study explored the relationship between fear of COVID-19, health anxiety, and social relations in WLHIV via path analysis. Among variables with significant causal relationships with social relations, SES showed the greatest negative relationship, and health anxiety demonstrated the greatest positive relationship on the direct path. Fear of COVID-19 showed the greatest positive relationship on the indirect path. Level of education was the only variable that had significant and positive relationships with social relations on both direct and indirect paths.

SES had the highest negative relationship with social relations; the poorer the socioeconomic status, the higher the social relations. Social relations means any relationship between two or more individuals, such as emotional and practical support provided by significant others [11]; in a special situation such as a pandemic, people with high SES decrease their relationship with the community and keep their distance. On the contrary, other studies reported fewer social relations in lower SES groups [20]. In another study, patients with HIV who had a lower SES were poorer or had lower living standards and had fewer social relations than others [21]. This difference in results can be attributed to the pandemic and differences among countries, as in our study, those with a higher SES were less in need of being in society to earn a living or visit healthcare centres. These factors limit social relations, especially for WLHIV.

**Table 4. Goodnees of fit indecesof empirical path model between health anxiety, COVID-19 phobia, and social relations.**

| Fit Index | $X^2$ | df | $X^2$/df | CFI | GFI | NFI | RMSEA |
|---|---|---|---|---|---|---|---|
| Model Index | 18.47 | 8 | 2.308 | 0.97 | 0.98 | 0.095 | 0.048 |
| Acceptablerange | X2/df < 5 | | | > 0.9 | > 0.9 | > 0.9 | < 0.05 |

CFI (comparative fit index), GFI (Goodness of fit index), NFI (Bentler-Bonett Normed fit index), RMSEA (root mean squared error of approximation)

Our study revealed that health anxiety had the greatest positive relationship with social relations. The main element of health anxiety is worry about health and the fear of disease. Commonly, the symptoms of worry and anxiety, which include many bodily symptoms, are misinterpreted as evidence of organic illness [22]. It causes great suffering for patients and those around them and is costly in terms of greater use of medical care utilization. For example, individuals suffering from health anxiety desperately seek to identify the physical causes of their symptoms and will often consult several medical professionals. Evidence suggests that the social costs of health anxiety are high. In undergraduates, health anxiety is linked to increased doctor visits [23]. We found no similar study on patients with HIV regarding this subject.

Nonetheless, according to previous studies, a possible cause of this finding could be that patients with more health anxiety visit healthcare centers more than others to check their health status. In Covid -19 pandemic, to ensure the diagnosis, they visit different doctors to make sure they are not infected with COVID-19 [9, 24], Which in turn may increase their social relations. The physical signs and symptoms of health anxiety during the pandemic may resemble the signs and symptoms of COVID-19 itself; in this case, people may mistake these physical changes as symptoms of COVID-19. People with high health anxiety regard any physical change as a sign of a disease, which exacerbates their anxiety and concern, and leads to repeated referrals [24].

People with severe immune deficiencies, such as HIV, face numerous side effects. They may be exposed to severe COVID-19 and have a higher mortality risk due to its complications, which can cause or exacerbate their stress and concern [25].

In the present study, fear of COVID-19 had the strongest positive relationship with social relations through the indirect path. Fear of COVID-19 positively affected health anxiety and thus increased the social relations of the patients. As noted before, WLHIV frequently visit diagnostic and treatment centers due to concerns and fear of COVID-19 complications, which increases their social relations [9, 24]. According to COVID-19 research and the media report, fear of COVID-19 affected has increased and fear of infection has become a concern in the context of the COVID-19 pandemic because it worsens emotion, cognition, and behavioral responses [26]. Although fear is a common psychological outcome during the pandemic, it is not limited to morbidity and mortality but may also emerge as social and occupational stress due to the evolving nature of the disease, its prevalence, and its unique risk factors. COVID-19 phobia is a hyper-reactive fear of contracting COVID-19 with three physiologic, cognitive, and behavioural components. Ongoing worry can induce symptoms such as tachycardia, tremor, breathing difficulty, vertigo, a changed appetite, obsession, and affective responses (sadness, guilt, anger). To prevent the consequences, people adopt avoidant behaviours that may disrupt the overall quality of their daily functioning [27]. Studies show that the complications and mortality caused by this disease are higher in people with chronic diseases, which induces or exacerbates fear and anxiety in them [28, 29].

In the current study, level of education was the only variable that had significant and positive relationships with social relations on both direct and indirect paths. Likewise, Nojoomi et al. showed that HIV-positive patients who were educated and employed had a better status than other patients in most quality of life dimensions, especially mental health, social functioning, and environmental dimension [30]. Educated patients have a better attitude towards the disease and are better adjusted to it due to their better occupational and financial opportunities and high cultural status, which can expand their social relations, such as emotional and practical support provided by significant others and leads to a better quality of life. Studies show that people with a higher level of education run a longer and healthier life compared to people of the same age but with a lower level of education [11, 31].

Contrary to our study, a study in China demonstrated no significant relationship between the social relations of patients with HIV and their level of education [32]. Cultural differences, economic conditions, and living standards could explain these differences.

This study had some limitations. One of them was the time in which the survey was conducted (during the COVID-19 pandemic); when adhering to social distancing and limiting social relations was necessary. In addition, this study didn't evaluate the patients' comorbidities, and used questionnaires for data recording. Nonetheless, due to the use of Path analysis based on multiple regression techniques, all confounders were adjusted so that they would not affect the reults.

## 5. Conclusion

According to our result, increased fear, health anxiety score and the duration of HIV, related to a higher score of social relations. Based on their vulnerability, WLHIV requires more support and need proper education to adhere to health protocols in future pandemics and similar situations.

## Supporting information

**S1 Checklist. STROBE statement—Checklist of items that should be included in reports of *cross-sectional studies*.**
(DOCX)

## Acknowledgments

Authors would like to appreciate the Honorable Vice-Chancellor of Research in Alborz University of Medical Sciences, Iranian research center for HIV/AIDS of Tehran University of Medical Sciences due to their scientific support as well as the staff of behavioral disease counselling center of Imam Khomeini hospital and all those who participated in this study.

## Author Contributions

**Conceptualization:** Fatemeh Aliverdi, Mostafa Qorbani, Zohreh Mahmoodi.

**Data curation:** Fatemeh Aliverdi, Zohreh Mahmoodi.

**Formal analysis:** Mostafa Qorbani, Zohreh Mahmoodi.

**Methodology:** Nooshin Ghavidel, Mostafa Qorbani, Farima Mohamadi, Zohreh Mahmoodi.

**Resources:** Zahra Bayat Jozani, Minoo Mohraz.

**Supervision:** Nooshin Ghavidel, Nami Mohammadian Khonsari, Farima Mohamadi, Zohreh Mahmoodi.

**Validation:** Minoo Mohraz.

**Writing – original draft:** Nami Mohammadian Khonsari.

**Writing – review & editing:** Fatemeh Aliverdi, Zahra Bayat Jozani, Nooshin Ghavidel, Nami Mohammadian Khonsari.

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
