## [Decision Letter · Decision Letter 0]

23 May 2022

PONE-D-22-08845Relationships among Corona phobia, health anxiety, and social relations in people living with HIV (PLHIV): A path analysisPLOS ONE

Dear Dr. Qorbani,

Thank you for submitting your manuscript to PLOS ONE. After careful consideration, we feel that it has merit but does not fully meet PLOS ONE’s publication criteria as it currently stands. Therefore, we invite you to submit a revised version of the manuscript that addresses the points raised during the review process.

We look forward to receiving your revised manuscript.

Kind regards,

Alok Atreya

Academic Editor

PLOS ONE

Journal Requirements:

2. During your revisions, please note that a simple title correction is required: change "Corona" to "COVID-19". Please ensure this is updated in the manuscript file and the online submission information.

Reviewers' comments:

Reviewer's Responses to Questions

**Comments to the Author**

1. Is the manuscript technically sound, and do the data support the conclusions?

Reviewer #1: No

Reviewer #2: Yes

2. Has the statistical analysis been performed appropriately and rigorously? 

Reviewer #1: No

Reviewer #2: Yes

3. Have the authors made all data underlying the findings in their manuscript fully available?

Reviewer #1: No

Reviewer #2: Yes

4. Is the manuscript presented in an intelligible fashion and written in standard English?

Reviewer #1: No

Reviewer #2: Yes

5. Review Comments to the Author

Reviewer #1: This study examined the interrelationships of socioeconomic characteristics, fear against COVID-19, health anxiety with social relations among people living with HIV. This manuscript highlights an important public health topic. I hope the following comments clarify the areas of improvement in the reporting of this study.

1. The title and abstract do not have the information on the study site. In the Methods section of the Abstract has the description that “enrolled 300 PLHIV who had records at the Behavioral Diseases Center of Tehran University of Medical Sciences,” though. The international audience may not understand where this study was conducted.

2. Citation format is inconsistent with the authors’ guidelines.

3. Study design: What is a “descriptive-analytical study?” Epidemiologists often distinguish clearly between a descriptive study and a analytical study. Broadly speaking they regard that a descriptive study as a means of exploring a hypothesis and an analytical study as for testing a hypothesis (for example, see the following: United States Centers for Disease Control and Prevention. Principles of epidemiology in public health practice. 3rd edition: An introduction to applied epidemiology and biostatistics. Lesson 1, Chapters 6 and 7. Atlanta: U.S. Department of Health and Human Services; 2012). The authors may want to clarify the study design based on what is intended to do in this study.

4. Items of the scale: Readers cannot obtain explanations on the questionnaire items regarding fear of COVID-19, anxiety, and social relations. Particularly, social relations may have different aspects by “relations with whom.” The authors need to explain more about the items in these scales.

5. Table 2: It would be good to have social relation at the right end of the column so that readers can check the correlation coefficients of different variable with social relation (as an outcome variable according to the objective statement at the end of the Background section) vertically in the single column.

6. Correlation analysis: It is not clear how the authors used the results of Table 2. How is it meaningful to how the size and significance of correlation coefficients? The authors may note that a significant correlation with social relations indicates that it is likely that the correlation coefficient is not zero in the hypothesis testing, which does not imply that a marginal change of a variable had a substantial increase in social relations score. Among variables examined in Table 2, HIV duration does not appear in subsequent path analysis, although HIV duration was significantly correlated with social relation. The authors need to explain how the correlation analysis was used in this study.

7. Path analysis: Does path analysis can be used to infer a causal relationship between variables? The authors may want to add reference to a theoretical literature that guarantees that. Otherwise, the authors may want to remove all the descriptions that imply causal relationships from the entire manuscript.

8. Table 3: Should the total effect be the summation of the indirect and direct effects, even though they are not statistically significant? If so, please reexamine all the coefficients in Table 3 to ensure if they are correct. In addition,

9. Figure 1: It would be nicer to indicate if each of the coefficients was significant or not in Figure 1. Significance in the paths to social relations was presented in Table 3, although other paths do not have information on significance. It is important to interpret which indirect paths were significant (for example, according to Table 3, an indirect effect of corona phobia was significant, although readers do not know through which paths corona phobia might affect social relations.

10. Discussion: As it is unclear what the variable of social relations measured, it is difficult to examine if the interpretations in the Discussion section are relevant. Particularly, it is difficult to consider how repetition of visiting health facilities and physical signs and symptoms are important in the association between health anxiety and social relations.

11. Limitations: The authors may want to list the limitations of this study and explain how these limitations affected the results of this study and the interpretation of these results at the end of the Discussion section.

Reviewer #2: Please see attachment for specific comments.

This is an interesting study. In general, the recommendations and the importance of the study are missing, so I am unable to understand the significance on a global scale. Also the limitations for interpretation are needed.

6. PLOS authors have the option to publish the peer review history of their article (what does this mean?). If published, this will include your full peer review and any attached files.

Reviewer #1: **Yes: **Akira Shibanuma

Reviewer #2: No

---

## [Author Response · Author response to Decision Letter 0]

22 Aug 2022

Dear Editor,

Thanks for your respectful comments. We correct all of them as follows:

Journal Requirements:

2. During your revisions, please note that a simple title correction is required: change "Corona" to "COVID-19". Please ensure this is updated in the manuscript file and the online submission information.

Answer: thanks for your comment .it was corrected and are highlighted in green.

Answer: thanks for your comment, we have added that “Accession numbers and/or DOIs will be made available after acceptance.” Hence if this manuscript was accepted, we will male the data publicaly available. 

Answer: thanks for your comment. We wrote the ethics statement in "2.4. Procedure of 2. Methods section".

Answer: thanks for your comment; all figures and tables were moved to the end of the manuscript. 

Thanks for the reviewer's respect and comments. We correct all of them as follows:

Reviewer #1: 

1. The title and abstract do not have the information on the study site. In the Methods section of the Abstract has the description that "enrolled 300 PLHIV who had records at the Behavioral Diseases Center of Tehran University of Medical Sciences," though. The international audience may not understand where this study was conducted.

Answer: thanks for your comment. 

We added a sentence in the background for more information with attention to word limitations.

Behavioral Diseases Center of Tehran University of Medical Sciences corrected as "Iranian research center for HIV/AIDS of Tehran University of Medical Sciences".

Moreover Iran was added to the title as well. Do illustrate the site of the study

2. Citation format is inconsistent with the authors' guidelines.

Answer: thanks for your comment. Citations were corrected.

3. Study design: What is a "descriptive-analytical study?" Epidemiologists often distinguish clearly between a descriptive study and a analytical study. Broadly speaking they regard that a descriptive study as a means of exploring a hypothesis and an analytical study as for testing a hypothesis (for example, see the following: United States Centers for Disease Control and Prevention. Principles of epidemiology in public health practice. 3rd edition: An introduction to applied epidemiology and biostatistics. Lesson 1, Chapters 6 and 7. Atlanta: U.S. Department of Health and Human Services; 2012). The authors may want to clarify the study design based on what is intended to do in this study.

Answer: thanks for your comment. We corrected it to cross-sectional study.

4. Items of the scale: Readers cannot obtain explanations on the questionnaire items regarding fear of COVID-19, anxiety, and social relations. Particularly, social relations may have different aspects by "relations with whom." The authors need to explain more about the items in these scales. 

Answer: thanks for your comment. The scale has been further elaborated.

5. Table 2: It would be good to have social relation at the right end of the column so that readers can check the correlation coefficients of different variable with social relation (as an outcome variable according to the objective statement at the end of the Background section) vertically in the single column.

Answer: thanks for your comment. These were implemented. 

6. Correlation analysis: It is not clear how the authors used the results of Table 2. How is it meaningful to how the size and significance of correlation coefficients? The authors may note that a significant correlation with social relations indicates that it is likely that the correlation coefficient is not zero in the hypothesis testing, which does not imply that a marginal change of a variable had a substantial increase in social relations score. Among variables examined in Table 2, HIV duration does not appear in subsequent path analysis, although HIV duration was significantly correlated with social relation. The authors need to explain how the correlation analysis was used in this study.

Answer: thanks for your attention. According to MUNRO'S Statistical Methods for Health Care Research book (2013) (1), one of the Statistical Assumptions for doing Path analysis is a correlation between variables. So we have to this analysis between variables

About HIV duration, it is correct. Excuse me. I analyzed it again, and we made all changes. If T-Value is under 1.96, the pathway isn't significant. So we put both images with T-value and standard B for better understanding.

 7. Path analysis: Does path analysis can be used to infer a causal relationship between variables? The authors may want to add reference to a theoretical literature that guarantees that. Otherwise, the authors may want to remove all the descriptions that imply causal relationships from the entire manuscript.

Answer: thanks for your attention. According to the majority of references, some of which are cited here, path analysis can be used as a statistical method to examine the effect of variables on one another, in other words, the causal relationship between variables, including:

MUNRO'S Statistical Methods for Health Care Research book (2013) explained that:

Path analysis is considered a causal modelling technique; it can be performed with either cross-sectional or longitudinal data. Path models are considered a type of causal model, and path analysis is referred to as a causal modelling technique. Path models depict theorized, directional relationships among a set of variables.

Path analysis is literally the analysis of the paths or lines in a model that represent the influence of one variable on another. It is used to answer research questions about the effect of a given independent (X1) variable on the dependent variable (Y) in the model. however, path analysis typically involves testing a causal or path model with data that do not result from an experimental Design, For example, path analysis can be done with survey data, data produced by a review of medical records, and so forth.(1)

B Shipley et al. (2016) suggested that some statistical methods can be used to investigate causal hypotheses and questions observational studies; one of these statistical methods is path analysis(2). 

Vieira A (2011) also stated that path analysis is a type of multiple regression statistical analysis used to examine causal models by examining the relationships between a dependent variable and two or more independent variables. We can use this method to estimate the extent and importance of causal relationships between variables. (3)

In other word Path analysis is a statistical technique that discern and assess the effects of a set of variables acting on a specified outcome via multiple causal pathways. This method allows users to investigate patterns of effect within a system of variables. It is one of several types of the general linear model that examine the impact of a set of predictor variables on multiple dependent variables. (3)

In this statistical method, as explained in the working method, path analysis is an extension of conventional regression that shows not only the direct effects but also the indirect effects of each variable on the dependent variables, and the results can be used to provide a rational interpretation of the relationships and correlations observed. Also, according to reviewer comments, we explained more and added a reference.

8. Table 3: Should the total effect be the summation of the indirect and direct effects, even though they are not statistically significant? If so, please reexamine all the coefficients in Table 3 to ensure if they are correct. 

Answer: thanks for your attention. The total effect is the summation of the indirect and direct effects, but in this way:

 If both, direct and indirect paths, are significant(according to T-value>1.96), the total effect, which is the summation of the indirect and direct effects, is significant too.

 If both direct/indirect pathways aren't significant, the total effect, which is the summation of direct and indirect paths, isn't significant.

If one path is significant, direct or indirect, that way we may not sum.

Nonrtheless, Table 3 and all its findings were re-assessed and evaluated 

9. Figure 1: It would be nicer to indicate if each of the coefficients was significant or not in Figure 1. Significance in the paths to social relations was presented in Table 3, although other paths do not have information on significance. It is important to interpret which indirect paths were significant (for example, according to Table 3, an indirect effect of corona phobia was significant, although readers do not know through which paths corona phobia might affect social relations.

Answer: thanks for your attention. According to the reviewer's comments, we put two figures; figure1 is according to T-value. If T-value is above 1.96, which indicates the significance of the paths (shown in red color). Figure 2 is the B standard of the paths

10. Discussion: As it is unclear what the variable of social relations measured, it is difficult to examine if the interpretations in the Discussion section are relevant. Particularly, it is difficult to consider how repetition of visiting health facilities and physical signs and symptoms are important in the association between health anxiety and social relations.

Answer: thanks for your attention. According to the reviewer's comments, we explained it in "2.3. Data collection and definition of terms." This questionnaire evaluates communication skills and social functioning with others/community and consists of 11 items like "I don't worry about dealing with people, I have competence and ability in social affairs, I am interested in joining a private group, I can be successful in verbal communication with others, Weak social connections can be due to failures caused by the living environment and society." And we added some references in the discussion part for more explanation.

11. Limitations: The authors may want to list the limitations of this study and explain how these limitations affected the results of this study and the interpretation of these results at the end of the Discussion section.

Answer: thanks for your attention. According to the reviewer's comments, we added it at the end of the discussion.

Reviewer #2: Please see attachment for specific comments.

This is an interesting study. In general, the recommendations and the importance of the study are missing, so I am unable to understand the significance on a global scale. Also the limitations for interpretation are needed.

Answer: thanks for your attention. According to the reviewer's comments, we added some references in the background, methods, and discussion parts. And we answered all of the questions in the attachment as follows respectively:

Q1=Does this mean staying away from or fear of going to a health facility?

Answer: thanks for your attention, yes. Because it was done during pandemic of covid

Q2=Which disease? COVID or HIV?

Answer: This group which live with HIV , it was substituted with PLHIV

Q3=Is this recognised in DSM?

Answer: Some theorists indicate that health anxiety would be better categorized as part of the OCD spectrum disorders and in DSM-5 added a new category of disorders called Obsessive-Compulsive and Related Disorders aswell (OCRDs) (also called Obsessive-Compulsive Spectrum Disorders in the research literature)

Q4=This contradicts earlier statements about avoidance of facilities.

Answer: these are a variety of theories about health anxiety.In both OCD and health anxiety individuals experience distressing intrusive repetitive thoughts that are difficult to resist and result in high levels of anxiety . it means some people with this disturbance have a fear of going to health centers and some people have an obsession to go to the health centers. This was added to the text as well.

Q5=What is social incompetence?

Answer: social incompetence , is the same as social disability. This was added to the text aswell.

Q6=Define social relations.

Also it its written in singular in some places and plural in others so make sure its consistent.

Answer: Thank you for your comments,we added its definition and corrected the errors

Q7,8,9=Relations?

Answer: thanks .corrected

Q10=How was the correlation calculated?

Answer: Thanks for your comment .as we wrote in samplesize ,it determined according to paper of Pizzirusso, M., Carrion-Park, C., Clark, U. S., Gonzalez, J., Byrd, D., & Morgello, S. (2021). Physical and mental health screening in a New York City HIV cohort during the COVID-19 pandemic: A preliminary

Q11=Please check the table because these scores are written with SD first in the table

Answer: Thanks for your attention. we corrected it.

Q12=Also possible that the relationship goes the other way - due to necessitated social relations, these people may have higher anxiety as the result of perceived risk and exposure?

Answer: Thanks for your comment. it is correct these can affect oneanother, but in the path analysis, is based on one way assessment and according to the result, health anxiety had an effect on social relations, so had to discuss our findings based on these results (this way), However we elaborated the paragraph further.

Q13= Might be useful to contrast this with information on health service utilisation to show whether this was the case?

Answer: Thanks for your comment.it is a great recommendation which we are preparing an other article to evaluate the above.

Q14=The rest of this paragraph does not tie in well. The reason why fear may result in increased social relations should be explored more.

Answer. thanks for your comments. we explained more as follow:According to COVID-19 research and the media report, fear of COVID-19 affected was increased. fear of infection has become a concern in the context of the COVID-19 pandemic because it worsens emotion, cognition, and behavioral responses(Quadros, Garg, Ranjan, Vijayasarathi, & Mamun, 2021)

Q15=This contradicts the earlier statement about increased care seeking

Answer:thanks for your comment. We added ” It causes great suffering for patients and those around them and is costly in terms of greater use of medical care utilization. For example, individuals suffering from health anxiety desperately seek to identify the physical causes of their symptoms and will often consult several medical professionals. Evidence suggests that the social costs of health anxiety are high. In undergraduates, health anxiety is linked to increased doctor visits. (Kobori, Okita, Shiraishi, Hasegawa, & Iyo, 2014)” their life affected by this situation and suffered which effect on their function ,quality of life and etc. to further elaborate what we ment, moreover based on the 4th comment we added that “some have a fear of going to health centers and some people have an obsession to go to the health centers. This was added to the text as well.” To avoid controversies

Q16=This part about financial status contradicts the earlier statement that increased SES led to decreased social relations, esp since education and SES are linked - how do you explain this?

Answer: Thanks for your comment. this part explained the effect of the education level on social relations. it is correct that education level is related to SES and good financial situation but as we explained about social relations (in instrument and in Background), it means that a person with a high level of education and a suitable SES situation, can use their relations, such as emotional and practical support provided by significant others, and stay home so they keep their distance from the community in this situation (Umberson & Karas Montez, 2010) We added some references.

Q17= The recommendations or areas for further consideration are a bit sparse. I am left wondering "So What" - what does one do with this info?

Answer: thanks for your attention we corrected and changed according to the title.

According to our result, increased fear, health anxiety score and duration of HIV affected, are related to a higher score of social relations. Based on their vulnerability, PLHIV requires more support and education to adhere to health protocols. This was added to the conclusion. 

Q18 This differs from text. Also write what the minimum and maximum score COULD be so that we know where in the range of possible scores the mean falls.

Answer: thanks for your comment .we corrected and we wrote the minimum and maximum score in result part.

Q19=Why are these qualitative variables - they are reported quantitatively?

Also the table is a bit confusing to read. Rather have all the variables in rows instead of two columns

Answer: thanks for your comment. we collect data in both forms, quantity and quality. For path analysis, because we have to use the quantity we use that form but for reporting for better understanding we report the quality form of them.

Q20= Why only women? Whats the sample of men and the distribution of men vs women?

Answer: thank you for the point. We corrected all terms in the manuscript. Since most male patients did not consent to participate, or they did not respond to us or were unreachable, unlike the women. We had no choice but to include the female patients.

references

1. Plichta SB, Kelvin EA, Munro BH. Munro's statistical methods for health care research: Wolters Kluwer Health/Lippincott Williams & Wilkins; 2013.

2. Shipley B. Cause and correlation in biology: a user's guide to path analysis, structural equations and causal inference with R: Cambridge University Press; 2016.

3. Jupp V. The Sage dictionary of social research methods: Sage; 2006.

---

## [Decision Letter · Decision Letter 1]

19 Sep 2022

Relationships among COVID-19 phobia, health anxiety, and social relations in women living with HIV in Iran: A path analysis

PONE-D-22-08845R1

Dear Dr. Qorbani,

We’re pleased to inform you that your manuscript has been judged scientifically suitable for publication and will be formally accepted for publication once it meets all outstanding technical requirements.

Kind regards,

Remya Lathabhavan

Academic Editor

PLOS ONE

Additional Editor Comments (optional):

Reviewers' comments:

Reviewer's Responses to Questions

**Comments to the Author**

1. If the authors have adequately addressed your comments raised in a previous round of review and you feel that this manuscript is now acceptable for publication, you may indicate that here to bypass the “Comments to the Author” section, enter your conflict of interest statement in the “Confidential to Editor” section, and submit your "Accept" recommendation.

Reviewer #2: All comments have been addressed

Reviewer #3: All comments have been addressed

2. Is the manuscript technically sound, and do the data support the conclusions?

Reviewer #2: Yes

Reviewer #3: Yes

3. Has the statistical analysis been performed appropriately and rigorously? 

Reviewer #2: Yes

Reviewer #3: Yes

4. Have the authors made all data underlying the findings in their manuscript fully available?

Reviewer #2: Yes

Reviewer #3: Yes

5. Is the manuscript presented in an intelligible fashion and written in standard English?

Reviewer #2: Yes

Reviewer #3: Yes

6. Review Comments to the Author

Reviewer #2: All comments have been addressed adequately in the response and/or in the text. The authors have edited the text to either clarify areas where ambiguity was present or to include supporting information.

Reviewer #3: The content of the paper is valuable and reflects great efforts which are much appreciated. The manuscript seems technically sound and the data support the conclusion.

7. PLOS authors have the option to publish the peer review history of their article (what does this mean?). If published, this will include your full peer review and any attached files.

Reviewer #2: No

Reviewer #3: No

---

## [Editor Report · Acceptance letter]

23 Sep 2022

PONE-D-22-08845R1 

Relationships among COVID-19 phobia, health anxiety, and social relations in women living with HIV in Iran: A path analysis 

Dear Dr. Qorbani:

I'm pleased to inform you that your manuscript has been deemed suitable for publication in PLOS ONE. Congratulations! Your manuscript is now with our production department. 

Kind regards, 

on behalf of

Dr. Remya Lathabhavan 

Academic Editor

PLOS ONE